

# Exogenous Hemin enhances the antioxidant defense system of rice by regulating the AsA-GSH cycle under NaCl stress

Fengyan Meng[1,2], Naijie Feng[1,2,3], Dianfeng Zheng[1,2,3], Meiling Liu[1,2], Hang Zhou[2,3], Rongjun Zhang[1,2], XiXin Huang[1,2] and Anqi Huang[1,2]

[1] College of Coastal Agriculture Sciences, Guangdong Ocean University, Zhanjiang, Guangdong, China
[2] South China Center of National Saline-tolerant Rice Technology Innovation Cente, Zhanjiang, Guangdong, China
[3] Shenzhen Research Institute of Guangdong Ocean University, Shenzhen, Guangdong, China

## ABSTRACT

Abiotic stress caused by soil salinization remains a major global challenge that threatens and severely impacts crop growth, causing yield reduction worldwide. In this study, we aim to investigate the damage of salt stress on the leaf physiology of two varieties of rice (Huanghuazhan, HHZ, and Xiangliangyou900, XLY900) and the regulatory mechanism of Hemin to maintain seedling growth under the imposed stress. Rice leaves were sprayed with 5.0 $\mu mol \cdot L^{-1}$ Hemin or 25.0 $\mu mol \cdot L^{-1}$ ZnPP (Zinc protoporphyrin IX) at the three leaf and one heart stage, followed by an imposed salt stress treatment regime (50.0 $mmol \cdot L^{-1}$ sodium chloride (NaCl)). The findings revealed that NaCl stress increased antioxidant enzymes activities and decreased the content of nonenzymatic antioxidants such as ascorbate (AsA) and glutathione (GSH). Furthermore, the content of osmoregulatory substances like soluble proteins and proline was raised. Moreover, salt stress increased reactive oxygen species (ROS) content in the leaves of the two varieties. However, spraying with Hemin increased the activities of antioxidants such as superoxide dismutase (SOD), peroxidase (POD), and catalase (CAT) and accelerated AsA-GSH cycling to remove excess ROS. In summary, Hemin reduced the effect of salt stress on the physiological characteristics of rice leaves due to improved antioxidant defense mechanisms that impeded lipid peroxidation. Thus, Hemin was demonstrated to lessen the damage caused by salt stress.

# INTRODUCTION

In the background of global warming, soil salinization has accelerated due to various factors such as seawater back-up, over-exploitation of groundwater, and the overdevelopment of arable land (*Alkharabsheh et al., 2021*). Saline land accounts for about one-fifth of the cultivated land and one-third of the irrigated farmland on the planet, and the area is increasing even faster (*Mukhopadhyay et al., 2021*). Salinity stress is one of the

Corresponding authors
Naijie Feng, fengnj@gdou.edu.cn
Dianfeng Zheng, zhengdf@gdou.edu.cn

most widespread and severe abiotic stresses globally. It has destructive effects on plant growth and physiological and biochemical processes and causes a decrease in grain production. According to current data, the yield loss caused by salt stress accounts for about 20.0% of global yield (*Ding et al., 2021*).

With salt stress increasing soil osmotic pressure, plant roots fail to absorb water and nutrients, which causes delayed growth and development or even death (*Liu et al., 2022*). In addition, salt stress induces excessive production of reactive oxygen species (ROS) in plant cells. ROS is weakly stable and easily causes oxidative stress to cells. The excessive ROS enhances cell membrane lipid peroxidation and disrupts membrane system stability, which results in the expansion of cell membrane permeability and extravasation of intracellular materials (*Seleiman et al., 2020*; *Hasnuzzaman et al., 2020*). It has been shown that ROS could break down proteins, damage DNA structure, and cause lipid peroxidation (*Chandrakar et al., 2017*; *Lin et al., 2020*). This disturbs the normal growth and physiological metabolic activities of plants. To avoid ROS accumulation, the plant will use antioxidant enzymes and non-enzymatic antioxidants to scavenge excess ROS (*Alisofi, Einalia & Sangtarash, 2020*). Among them, antioxidant enzymes include superoxide dismutase (SOD), peroxidase (POD), catalase (CAT), and ascorbate peroxidase (APX). Non-enzymatic antioxidants include ascorbic acid (AsA) and glutathione (GSH), which act as co-factors for different enzymes and participate in various metabolic processes (*Hasnuzzaman et al., 2020*). In addition, plants accumulate osmoregulatory substances to maintain internal and external cell balance *via* osmosis. There are two categories of osmoregulatory substances: inorganic ions ($Na^+$ and $K^+$) and organic substances, including proline and soluble proteins (*Athar et al., 2022*). Under salt stress, plants balance the osmotic pressure between the plant and the external environment by conducting selective uptake of ions and promoting the accumulation of phase-soluble solutes.

Rice (*Oryza sativa* L), a gramineous crop, has a long history of cultivation and consumption in China. The consumer demand for rice in China is the highest in the world, and more than half of the population consumes rice as their major food source (*Huang et al., 2022*; *Zuo et al., 2022*). However, salt stress has become one of the major abiotic stresses which limit rice production. Many studies showed that the seedling stage was an essential stage of rice development and was closely related to the later development of tillers and spikelets. However, this stage is susceptible to the impact of salt stress (*Zeng, Shannon & Lesch, 2001*). Therefore, it has become a research priority in recent years to find effective ways for improving the salt tolerance of rice seedlings.

Plant growth regulators (PGRs) are a group of synthetic compounds with phytohormonal activity that improve the tolerance to abiotic stresses by affecting the expression of endogenous hormones in crops. Hemin is a small molecule with a porphyrin structure, consisting of nitrogen atoms on four pyrrole rings in a porphyrin ligated to a ferrous ion. In recent years, Hemin has been used more frequently in different crops for its natural, non-polluting, low-cost, and high-safety features. Hemin acts as a substrate and promoter of heme oxygenase 1 (HO-1), an initiator and rate-limiting enzyme for Hemin degradation, which has a specific inhibitor, zinc protoporphyrin (ZnPP). Hemin triggered salt acclimation in wheat by increasing HO-1 expression, while ZnPP, an inhibitor, was

shown to decrease the salt tolerance of wheat (*Xie et al., 2011*). Under salt stress, Hemin increased proline and soluble protein contents, enhanced antioxidant enzyme activities such as SOD, CAT, and APX, and alleviated oxidative damage in *Cassia obtusifolia* L (sickle senna) (*Zhang et al., 2012*). In addition, under zinc (Zn), lead (Pb), and chromium (Cr) metal stress, Hemin activated the activities of various antioxidant enzymes (SOD, glutathione reductase (GR), and APX) in rice seedlings, improved the content of AsA and GSH, and reduced heavy metal accumulation.

At present, only a small number of studies have been conducted on the mitigation of salt stress by Hemin on rice seedings. Hemin has been more frequently used to mitigate other abiotic stresses in other plant species, such as heavy metal stress of *Medicago sativa* L (alfalfa) (*Fu et al., 2011*), low-temperature stress of *Conyza blini* (bear gall grass) (*Zheng et al., 2021*), and salt stress in *Brassica juncea* L (mustard) (*Verma et al., 2015*). Furthermore, spraying plant growth regulators can improve the resistance of rice seedlings during the critical period before transplanting, which is essential for the subsequent transplanting of rice seedlings on saline land. Hence, in this study, we used two rice varieties, Huanghuazhan and Xiangliangyou900, to research the impacts of Hemin on the growth and ROS metabolism (antioxidant enzymes and non-enzymatic antioxidants) of rice seedlings at the three leaf and one heart stage under salt stress. This experiment aims to reveal the mechanism of Hemin in enhancing the salt tolerance of rice and to provide theoretical basis and technical guidance for the cultivation of rice on saline soils.

## MATERIALS AND METHODS

### Plant materials

The experiment was conducted in 2022 at College of Coastal Agriculture Sciences in Guangdong Ocean University. To ensure broad coverage through our experimentation, we selected the conventional rice variety Huanghuazhan (HHZ) and the hybrid rice variety Xiangliangyou900 (XLY900). The Shanghai Zhangdeduo Agricultural Technology Co., Ltd provided Hemin.

### Experiment design

Seeds with uniformity of size and color were sterilized with 3.0% $H_2O_2$ for about 15.0 min, and then rinsed 3–5 times with distilled water. These seeds were soaked and germinated for 24 h under dark conditions at 30.0 °C. Sixty-five seeds were sown into pots containing 3.0 kg of test soil with 1:3 sand to latosol content. The plastic pot sizes were 19.0 cm for the upper diameter, 14.0 cm for the lower diameter, and 17.0 cm for the height, without holes at the bottom. Regular water irrigation was performed until the three leaf and one heart stage (about 18 days after planting). Rice leaves were sprayed with 5.0 µmol·L$^{-1}$ Hemin and 25.0 µmol·L$^{-1}$ ZnPP alone or in combination, and plants were exposed to 25.0 mmol·L$^{-1}$ NaCl stress twice at two 24 h intervals, which resulted in the salt concentration in the soil reaching 50.0 mmol·L$^{-1}$ at 48 h after spraying. In subsequent experiments, concentrations were maintained by measuring soil conductivity (EC = 5.0 ± 0.5 dS·m$^{-1}$). Each variety had five treatments: (1) normal water (CK); (2) 50.0 mmol·L$^{-1}$ NaCl (S); (3) Hemin + 50.0 mmol·L$^{-1}$ NaCl (SH); (4) ZnPP + 50.0 mmol·L$^{-1}$ NaCl (SZ); and (5) Hemin + ZnPP + 50.0

mmol·L$^{-1}$ NaCl (SZH). Each treatment had twenty-five pots. The plant samples were harvested on days 3, 5, 7, and 9 after NaCl stress application for morphological and physiological parameter assessment.

## Morphological measurements

Plant height was measured with a ruler; stem diameter was measured with a vernier; shoot fresh and dry weight were measured by a caliper electronic analytical balance. The shoots were dried for 30.0 min at 105.0 °C and for 72 h at 85.0 °C.

## Measurement of electrolyte leakage (EL), malonaldehyde (MDA), and Hydrogen peroxide (H$_2$O$_2$) content

EL was determined as described by *Yu et al. (2021)*. The measurement of MDA content was determined according to the method outlined by *Ahmad et al. (2016)*. In brief, the frozen leaf sample (0.5 g) was extracted in 10.0 mL phosphate buffer solution (PBS, 0.05 mmol·L$^{-1}$, pH 7.8) and centrifuged at 6,000 rpm for 20.0 min. One milliliter of the supernatant was added to 2.0 mL 2-thiobarbituric acid (0.6%, TBA), then boiled at 100.0 °C for 15.0 min. The mixture was cooled quickly with cold water and centrifuged at 4,000 rpm for 20.0 min. The absorption value was determined at 450 nm, 532 nm, and 600 nm. The H$_2$O$_2$ content was determined according to *Rasheed et al. (2022)*. Specifically, 0.5 g of the frozen sample was ground into homogenate in 5.0 mL of 0.1% trichloroacetic acid (TCA) and centrifuged at 19,000 rpm for 20.0 min. A 500 microliters of supernatant were added to 0.5 mL PBS (10.0 mmol·L$^{-1}$, pH 7.0) and 1.0 mL potassium iodide (1.0 mol·L$^{-1}$, KI), then the reaction mixture was incubated at 28.0 °C for 1.0 h in the dark. The absorbance values were recorded at 410 nm.

## Histochemical detection of hydrogen peroxide and superoxide anion

The histochemical staining of H$_2$O$_2$ and superoxide radicle (O$_2$·$^-$) was determined by the methods outlined in *Zhang et al. (2009)* and *Sudhakar et al. (2015)*, respectively. In brief, on day three of the stress application, the second leaf of CK, S, SH, SZ, and SZH treatments of both varieties were sampled and placed in a solution containing nitrogen blue tetrazolium (NBT) and 3,3′-diaminobenzidine (DAB) for staining. The leaves were vacuum infiltrated and then kept at room temperature and dark conditions for 24 h until brown and blue spots appeared, respectively. The staining solution was discarded. Ethanol (95.0%) was used to extract the chlorophyll in an 80.0 °C water bath. Ethanol was added continuously until the chlorophyll had been completely cleared from the leaf samples, and then these samples were used for photography.

## Measurement of the activities of superoxide dismutase (SOD), peroxidase (POD), and catalase (CAT)

The frozen leaf samples (0.5 g) were extracted in 10.0 mL PBS (50.0 mmol·L$^{-1}$, pH 7.8) centrifuged at 12,000 rpm at 4.0 °C for 20.0 min. The supernatant was used to determine SOD (EC 1.15.1.1), POD (EC 1.11.1.7), and CAT (EC 1.11.1.6) (*Habib et al., 2021*) activities. SOD activity was determined according to the method by *Lu et al. (2022)*.

The supernatant was mixed with 14.5 mmol·L$^{-1}$ methionine (Met) solution, 3.0 mmol·L$^{-1}$ EDTA-Na$_2$ solution, 60.0 μmol·L$^{-1}$ riboflavin solution, and 2.25 mmol·L$^{-1}$ nitrogen blue tetrazolium (NBT) solution. One unit of SOD activity was defined as the amount of enzyme that would inhibit 50.0% of NBT photoreduction. POD was determined following the method outlined by *Kenawy et al. (2022)*. The supernatant was mixed with PBS (pH 6.0), guaiacol, and 30.0% H$_2$O$_2$. The absorbance was measured at 470 nm. CAT was determined by the decreased absorbance rate of H$_2$O$_2$ at 240 nm, as described by *Basilio-Apolinar et al. (2021)*.

## Measurement of AsA-GSH cycle products and substrate content

The procedure outlined by *Costa, Gallego & Tomaro (2002)* and *Yan et al. (2021)* was followed to measure the contents of AsA and total AsA. Specifically, 0.5 g of frozen leaf sample was extracted in 5.0% trichloroacetic acid (TCA) and centrifuged at 12,000 rpm at 4.0 °C for 15.0 min. The supernatant was used to determine the content of AsA and total AsA. For AsA, the supernatant was mixed with a reaction solution containing 5.0% TCA, ethanol, 0.4% phosphoric acid (H$_3$PO$_4$)-ethanol, 0.5% bathophenanthroline (BP)-ethanol, and 0.03% ferric chloride (FeCl$_3$)-ethanol. The reaction was carried out at 30.0°C for 90.0 min. The absorbance was assayed at 534 nm. For total AsA, it was similar to the AsA assay. However, the sample solutions were first reacted with 60.0 mmol·L$^{-1}$ dithiothreitol (DTT)-ethanol solution and Na$_2$HPO$_4$ (0.2 mol·L$^{-1}$)–NaOH (1.2 mol·L$^{-1}$) solution for 10.0 min. Then, 20.0% TCA was added and mixed with the above reaction solution. The absorbance was assayed at 540 nm. Dehydroascorbate (DHA) content was calculated based on the difference between total AsA and reduced AsA.

The glutathione (GSH) and oxidized glutathione (GSSG) content was determined according to the method described by *Kaya et al. (2023)*. Namely, 0.5 g of frozen sample was ground into homogenate in 5.0 mL of 5.0% metaphosphoric acid (HPO$_3$) and centrifuged at 20,000 × *g* for 20.0 min. The supernatant was used to determine the content of total glutathione (GSH + GSSG) and oxidized glutathione (GSSG). The supernatant was mixed homogeneously with the reaction solution, which contained 5.0% sulfosalicylic acid, 1.84 mol·L$^{-1}$ triethanolamine (TEA), and was incubated in a 25.0 °C water bath for 1.0 h. Then 50.0 mmol·L$^{-1}$ PBS, 10.0 mmol·L$^{-1}$ nicotinamide adenine dinucleotide phosphate (NADPH), 12.5 mmol·L$^{-1}$ 5,5′-Dithiobis-(2-nitrobenzoic acid) (DTNB) were added, and the reaction was maintained at 25.0 °C for 10.0 min. Then, 50 U glutathione reductase (GR) was added to the reaction mixture. The absorbance value of (GSH + GSSG) was measured at 412 nm. Besides adding the reaction solution, which contained 5.0% sulfosalicylic acid, 1.84 mol·L$^{-1}$ TEA, and 2-vinyl pyridine (2-VP), the subsequent steps were kept consistent with the determination of (GSH + GSSG) content. The GSSG absorbance value was measured at 412 nm. GSH content was calculated according to the following formula.

The GSH content = GSH + GSSG content − GSSG content.
## Measurement of the critical enzyme indexes of the AsA-GSH cycle

Five hundred milligrams of the frozen leaf sample was placed in a mortar, ground into a powder with 5.0 mL PBS (50.0 mmol·L$^{-1}$, pH 7.8), and loaded into a centrifuge tube. The centrifuge tube was centrifuged at 12,000 × $g$ for 20.0 min. The supernatant was used to measure the levels of ascorbate peroxidase (APX) (EC 1.11.1.11), monodehydroascorbate reductase (MDHAR, EC 1.6.5.4), dehydroascorbate reductase (DHAR, EC 1.8.5.1) and glutathione reductase (GR, EC 1.6.4.2).

The APX activity was determined according to the method described by *Sharifi et al. (2021)*. The assay mixture contained 0.1 mL of the supernatant, 2.6 mL EDTA-Na$_2$ (0.1 mmol·L$^{-1}$), 0.15 mL AsA (5.0 mmol·L$^{-1}$) and 20.0 mmol·L$^{-1}$ H$_2$O$_2$. The absorbance was assayed at 290 nm. (E = 2.8 mM$^{-1}$ cm$^{-1}$). MDHAR activity was measured using the method described by *Hasanuzzaman, Hossain & Fujita (2011)*. The reaction mixture consisted of 25.0 mmol·L$^{-1}$ PBS (pH 7.8), 7.5 mmol·L$^{-1}$ AsA, 2.0 mmol·L$^{-1}$ NADPH, 50 U AsA oxidase (EC 1.10.3.3), and the supernatant. The absorbance was assayed at 340 nm. (E = 6.2 mM$^{-1}$ cm$^{-1}$). DHAR activity was determined using the method described by *Shan & Liu (2017)*. DHAR was assayed in a mixed solution containing 25.0 mmol·L$^{-1}$ PBS (pH 7.8), 20.0 mmol·L$^{-1}$ GSH, 10.0 mmol·L$^{-1}$ DHA, and the supernatant. The absorbance was assayed at 340 nm (E = 14 mM$^{-1}$ cm$^{-1}$). GR activity was determined according to *Keles & Oncel (2002)*. GR (EC 1.6.4.2) was assayed in a mixed solution containing 25.0 mmol·L$^{-1}$ PBS (pH 7.8), 2.0 mmol·L$^{-1}$ ethylene diamine tetra acetic acid (EDTA), 10.0 mmol·L$^{-1}$ GSSG, 24.0 mmol·L$^{-1}$ NADPH, and the supernatant. The absorbance was assayed at 340 nm (E = 6.2 mM$^{-1}$ cm$^{-1}$).

## Measurement of soluble protein and proline content

Soluble protein content was determined according to the method described by *Tian et al. (2022)*. The absorbance value was measured at 595 nm using Coomassie brilliant blue. Proline content was carried out according to the method by *Liu et al. (2020)*. The frozen sample (0.5 g) was ground in 5.0 mL of 3.0% sulfosalicylic acid and then centrifuged at 3,000 × $g$ for 10.0 min. Two milliliters of the supernatant were added to 2.0 mL acetic acid and 2.0 mL acidic ninhydrin and then incubated in a water bath at 100.0 °C for 30.0 min. After cooling, 4.0 mL of toluene was added, and the absorbance was measured at 520 nm.

## Statistical analysis

The data was analyzed using Microsoft Excel 2019 and SPSS 25.0. The figures were drawn in Origin 2021. Duncan test ($p < 0.05$) was used to evaluate the difference within treatments, and the significant differences among different materials were determined.

## RESULT

### The morphological parameters of rice seedlings

There was significant inhibition of rice growth under NaCl stress, which showed a remarkable decrease in plant height, stem base width, shoot fresh weight, and shoot dry weight (Tables 1 and 2). From days 3 to 9, in comparison to CK, the plant height, stem diameter, shoot fresh weight, and shoot dry weight of HHZ under NaCl stress significantly

**Table 1 Effects of exogenous Hemin on the morphological indexes of rice seedlings under NaCl stress.**

| Morphological indexes | Varieties | Treatments | NaCl stress time (d) | | | |
|---|---|---|---|---|---|---|
| | | | 3 | 5 | 7 | 9 |
| Plant height (cm) | HHZ | CK | 31.67 ± 0.20a | 32.63 ± 0.19a | 33.57 ± 0.07a | 33.63 ± 0.09a |
| | | S | 27.13 ± 0.71c | 27.93 ± 0.13d | 28.00 ± 0.00d | 29.10 ± 0.21d |
| | | SH | 29.83 ± 0.19b | 30.97 ± 0.17b | 31.47 ± 0.03b | 31.90 ± 0.15b |
| | | SZ | 27.00 ± 1.00c | 27.63 ± 0.57d | 27.90 ± 0.57d | 28.50 ± 0.32d |
| | | SZH | 28.93 ± 0.03b | 29.57 ± 0.03c | 29.80 ± 0.30c | 29.97 ± 0.09c |
| | XLY900 | CK | 32.90 ± 0.31a | 33.43 ± 0.18a | 33.80 ± 0.46a | 34.67 ± 0.03a |
| | | S | 28.30 ± 0.06d | 29.40 ± 0.12d | 29.70 ± 0.06d | 30.97 ± 0.27d |
| | | SH | 30.57 ± 0.03b | 30.97 ± 0.23b | 31.33 ± 0.18b | 33.17 ± 0.33b |
| | | SZ | 28.33 ± 0.03d | 29.50 ± 0.06d | 29.50 ± 0.06d | 30.50 ± 0.06d |
| | | SZH | 29.10 ± 0.10c | 30.25 ± 0.14c | 30.53 ± 0.03c | 32.00 ± 0.00c |
| Stem diameter (mm) | HHZ | CK | 3.27 ± 0.06a | 3.47 ± 0.06a | 3.63 ± 0.06a | 3.80 ± 0.10a |
| | | S | 2.43 ± 0.00e | 2.67 ± 0.12d | 2.63 ± 0.06e | 2.70 ± 0.12d |
| | | SH | 3.07 ± 0.06b | 3.20 ± 0.00b | 3.40 ± 0.00b | 3.57 ± 0.06b |
| | | SZ | 2.63 ± 0.15d | 2.67 ± 0.12d | 2.73 ± 0.06d | 2.87 ± 0.21d |
| | | SZH | 2.83 ± 0.06c | 2.93 ± 0.06c | 3.07 ± 0.06c | 3.20 ± 0.00c |
| | XLY900 | CK | 3.53 ± 0.03a | 3.63 ± 0.03a | 3.77 ± 0.03a | 3.90 ± 0.00a |
| | | S | 2.87 ± 0.03d | 3.00 ± 0.00d | 2.90 ± 0.06d | 3.00 ± 0.00d |
| | | SH | 3.37 ± 0.03b | 3.47 ± 0.09b | 3.60 ± 0.00b | 3.63 ± 0.03b |
| | | SZ | 2.93 ± 0.07d | 3.00 ± 0.06d | 2.93 ± 0.03d | 3.10 ± 0.06d |
| | | SZH | 3.20 ± 0.00c | 3.23 ± 0.03c | 3.40 ± 0.00c | 3.47 ± 0.03c |

**Note:**
Data in this table is mean ± standard error of at least three replicates. According to Duncan's multiple range tests, different lowercase letters indicate significant difference at the five percent significant level within each column.

decreased by 13.48–16.58%, 23.08–28.95%, 28.20–32.41%, and 21.14–23.34%, respectively. Similarly, in the XLY900 variety, the above indicators decreased by 10.67–13.98%, 17.43–23.08%, 27.24–30.71% and 18.22–22.15%, respectively. Exogenous Hemin alleviated the inhibition of rice seedling growth by NaCl stress (Fig. 1). From days 3 to 9, in comparison to the NaCl treatment, the plant height, stem diameter, shoot fresh weight and shoot dry weight of HHZ with SH treatment were significantly higher by 9.62–12.38%, 20.00–32.10%, 18.63–27.43%, and 11.96–15.84%, respectively. Similarly, in the XLY900 variety, the above indicators were increased by 5.33–8.01%, 15.56–24.14%, 15.85–26.58%, and 12.78–14.26%, respectively. This finding suggests that the Hemin effectively mitigated the detrimental effects of NaCl stress on rice seedling development. Hemin promoted a higher growth of HHZ seedlings. In contrast to the NaCl treatment, the ZnPP treatment did not increase plant height, stem base width, shoot fresh weight, or shoot dry weight in both rice variety. The addition of Hemin reversed the inhibition caused by ZnPP and enhanced the growth of rice seedlings. From days 3 to 9, in comparison to the SZ treatment, the plant height, stem diameter, shoot fresh weight and shoot dry weight of HHZ with SH treatment were increased by 5.15–7.16%, 7.59–12.20%, 9.12–19.43%, and 8.56–10.66%, respectively. Similarly, in the XLY900 variety, the plant height, stem

Table 2 Effects of exogenous Hemin on the biomass of rice seedlings under NaCl stress.

| Morphological indexes | Varieties | Treatments | NaCl stress time (d) | | | |
|---|---|---|---|---|---|---|
| | | | 3 | 5 | 7 | 9 |
| Shoot fresh weight (g) | HHZ | CK | 0.4771 ± 0.0060a | 0.4742 ± 0.0109a | 0.4829 ± 0.0009a | 0.5147 ± 0.0041a |
| | | S | 0.3225 ± 0.0092d | 0.3335 ± 0.0025d | 0.3467 ± 0.0043d | 0.3617 ± 0.0007d |
| | | SH | 0.4109 ± 0.0094b | 0.3956 ± 0.0029b | 0.4270 ± 0.0043b | 0.4430 ± 0.0029b |
| | | SZ | 0.3169 ± 0.0032d | 0.3399 ± 0.0056d | 0.3398 ± 0.0024d | 0.3610 ± 0.0091d |
| | | SZH | 0.3557 ± 0.0109c | 0.3709 ± 0.0040c | 0.4058 ± 0.0018c | 0.4180 ± 0.0068c |
| | XLY900 | CK | 0.5147 ± 0.0021a | 0.5110 ± 0.0262a | 0.5225 ± 0.0053a | 0.5477 ± 0.0098a |
| | | S | 0.3566 ± 0.0101d | 0.3577 ± 0.0068c | 0.3718 ± 0.0051c | 0.3985 ± 0.0014d |
| | | SH | 0.4385 ± 0.0076b | 0.4528 ± 0.0195b | 0.4546 ± 0.0318b | 0.4617 ± 0.0160b |
| | | SZ | 0.3623 ± 0.0033d | 0.3602 ± 0.0220c | 0.3732 ± 0.0132c | 0.3923 ± 0.0094d |
| | | SZH | 0.4034 ± 0.0057c | 0.4185 ± 0.0088b | 0.4260 ± 0.0016b | 0.4310 ± 0.0025c |
| Shoot dry weight (g) | HHZ | CK | 0.0938 ± 0.0014a | 0.0966 ± 0.0022a | 0.1004 ± 0.0007a | 0.1016 ± 0.0015a |
| | | S | 0.0740 ± 0.0023c | 0.0761 ± 0.0025c | 0.0770 ± 0.0007c | 0.0783 ± 0.0013d |
| | | SH | 0.0840 ± 0.0012b | 0.0852 ± 0.0013b | 0.0870 ± 0.0015b | 0.0907 ± 0.0013b |
| | | SZ | 0.0725 ± 0.0024c | 0.0751 ± 0.0004c | 0.0767 ± 0.0014c | 0.0779 ± 0.0011d |
| | | SZH | 0.0824 ± 0.0006b | 0.0831 ± 0.0015b | 0.0841 ± 0.0012b | 0.0846 ± 0.0015c |
| | XLY900 | CK | 0.0933 ± 0.0023a | 0.1016 ± 0.0017a | 0.1044 ± 0.0005a | 0.1096 ± 0.0045a |
| | | S | 0.0817 ± 0.0017d | 0.0825 ± 0.0015d | 0.0818 ± 0.0008cd | 0.0853 ± 0.0000cd |
| | | SH | 0.0933 ± 0.0006b | 0.0943 ± 0.0003b | 0.0934 ± 0.0029b | 0.0962 ± 0.0009b |
| | | SZ | 0.0804 ± 0.0023d | 0.0816 ± 0.0009d | 0.0802 ± 0.0043d | 0.0811 ± 0.0010d |
| | | SZH | 0.0876 ± 0.0006c | 0.0871 ± 0.0015c | 0.0880 ± 0.0000bc | 0.0895 ± 0.0006bc |

Note:
Data in this table is mean ± standard error of at least three replicates. According to Duncan's multiple range tests, different lowercase letters indicate significant difference at the five percent significant level within each column.

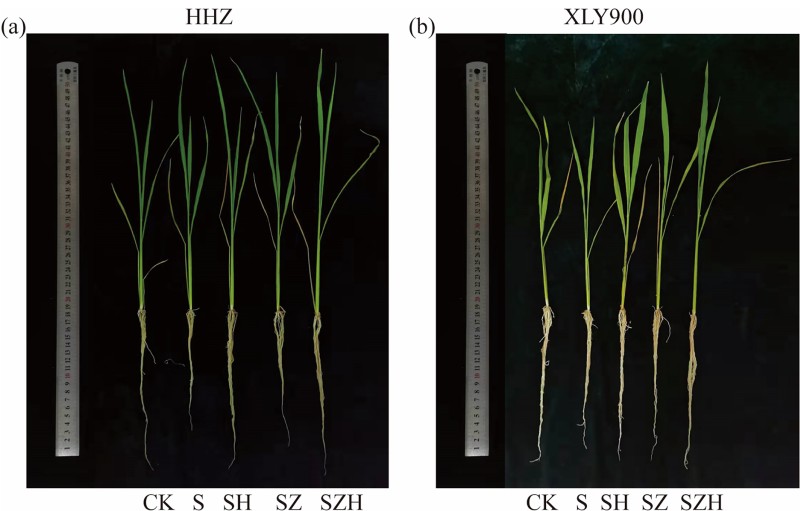

(a) HHZ     (b) XLY900

CK  S  SH  SZ  SZH          CK  S  SH  SZ  SZH

**Figure 1 Effect of Hemin on growth of rice seedlings under NaCl (on day 9) in HHZ (A) and XLY900 (B).**

diameter, shoot fresh weight and shoot dry weight with SH treatment were increased by 2.54–4.92%, 7.78–15.91%, 9.86–16.19%, and 6.70–10.32%, respectively.

## The membrane damage and ROS accumulation in rice leaves

Compared to the CK, EL, MDA, and $H_2O_2$ contents in the two assessed rice varieties gradually increased with the increased period of NaCl stress treatment (Fig. 2). Compared to CK, the EL of HHZ and XLY900 under NaCl stress significantly increased by 16.26–126.50% and 35.25–86.19% from days 3 through to 9, respectively. After NaCl stress, there was a significant rise in the MDA and $H_2O_2$ content of HHZ in the NaCl treatment. This increase ranged from 31.79% to 51.73% for MDA and 13.92% to 30.29% for $H_2O_2$ from days 3 to 9, compared to CK. In the NaCl treatment of XLY900, the contents of MDA and $H_2O_2$ were significantly increased by 22.25–40.52% and 20.26–25.09%, compared with CK, from days 3 to 9, respectively. The $H_2O_2$ and MDA contents of HHZ were higher than that of XLY900 on day 9 after NaCl stress, showing that NaCl stress was more harmful to HHZ, which was more sensitive to NaCl stress than the XLY900 variety.

Spraying Hemin effectively reduced EL and the content of MDA and $H_2O_2$ of both rice varieties compared with NaCl treatment. In contrast to the NaCl treatment, the EL of both HHZ and XLY900 exhibited a noticeable decrease in the SH treatment, including reductions of 9.64% to 28.20% and 8.78% to 18.41%, respectively. The MDA and $H_2O_2$ contents in the SH treatment of HHZ compared to the NaCl treatment decreased by 15.20–20.28% and 8.30–16.52%, respectively. Similarly, in the SH treatment of XLY900, MDA and $H_2O_2$ contents decreased by 11.59–18.15% and 5.97–15.72% compared to the NaCl treatment from days 3 to 9, respectively. EL, MDA, and $H_2O_2$ remained high in both varieties under ZnPP treatment. Throughout the stress period, the SZH treatment reduced EL, MDA contents, and $H_2O_2$ content of both HHZ and XLY900 compared to the SZ treatment. On days 3 and 9, compared to the SZ treatment, EL of HHZ exhibited noticeable decreased of 9.21% and 10.43%, respectively, in the SZH treatment. From days 3 to 9, compared to the SZ treatment, the EL of XLY900 with the SZH treatment declined by 6.09–9.01%. From days 3 to 9, compared with the SZ treatment, the content of MDA and $H_2O_2$ was decreased by 5.87–7.15% and 3.51–10.99% in HHZ with the SZH treatment, and were reduced by 1.44–7.71% and 1.22–9.71% in XLY900 with the SZH treatment, respectively.

## The histochemical localization of reactive oxygen species in rice leaves

The distribution of $H_2O_2$ and superoxide anion ($O_2 \cdot^-$) were localized and expressed visually by histochemical analysis of HHZ and XLY900 rice leaves. $H_2O_2$ was stained with dark brown spots, and $O_2 \cdot^-$ was stained with dark blue spots (Fig. 3). Compared to CK, dark brown and dark blue spots were significantly increased in rice leaves of both varieties under NaCl stress. Compared to the NaCl treatment, dark brown and dark blue spots on leaves were significantly decreased in abundance HHZ and XLY900 with the SH treatment, which indicated that foliar application of Hemin could potentially reduce the

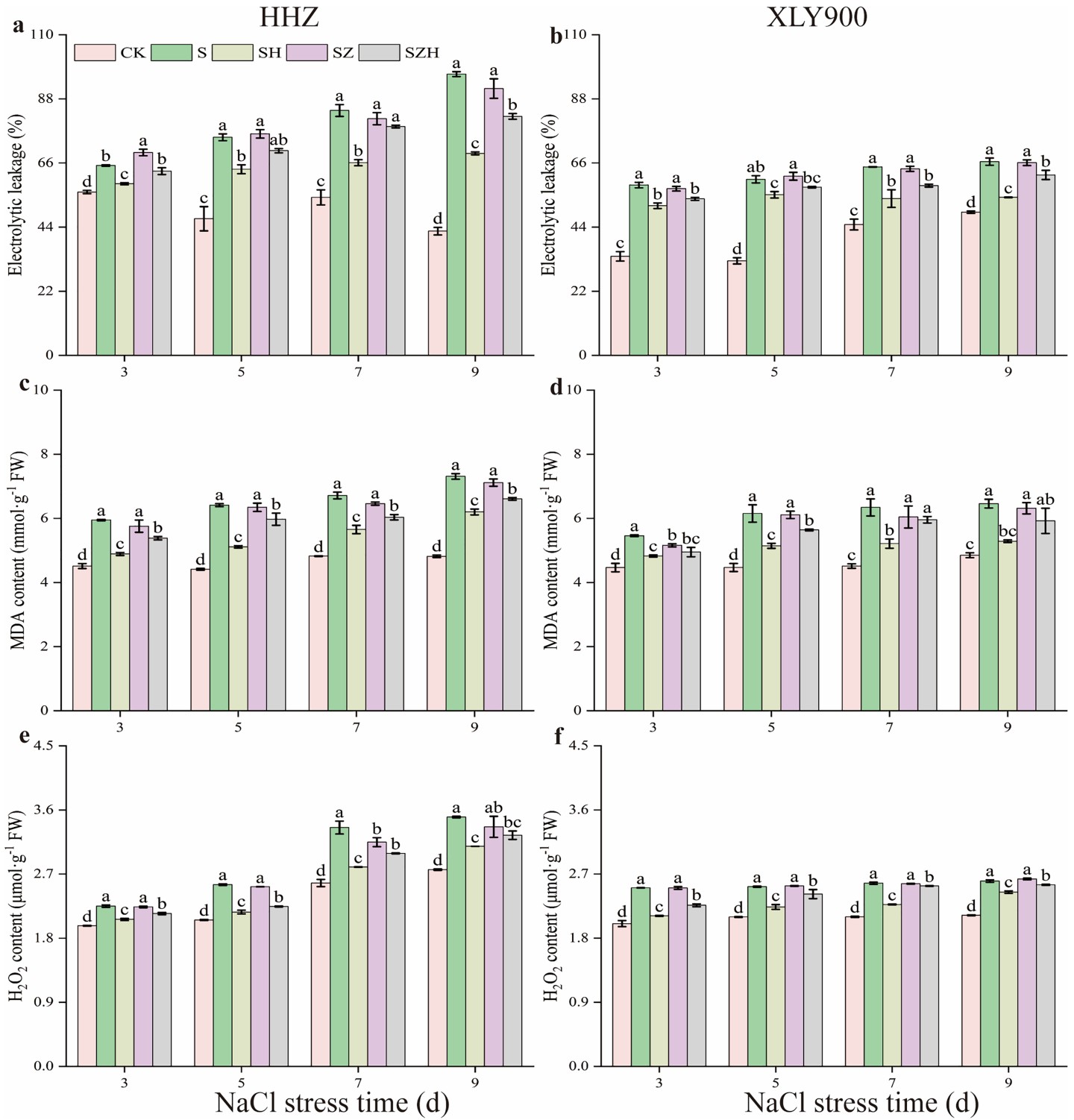

**Figure 2 Effect of Hemin on membrane damage and ROS accumulation of rice seedlings under NaCl.** Electrolyte leakage in HHZ (A) and XLY900 (B); MDA in HHZ (C) and XLY900 (D) and $H_2O_2$ in HHZ (E) and XLY900 (F). Values are the means ± SD of three replicate samples. Different lowercase letters in the data column indicate significant differences ($p < 0.05$) according to Duncan's test.

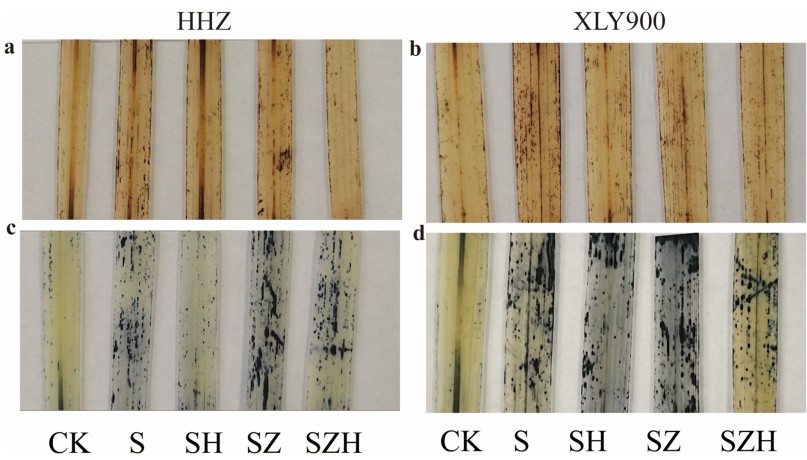

**Figure 3 Effect of Hemin on histochemical localization of H₂O₂ and O₂·⁻ on rice leaves under NaCl stress (on day 3).** $H_2O_2$ in HHZ (A) and XLY900 (B) and $O_2^{\cdot-}$ in HHZ (C) and XLY900 (D).

accumulation and distribution of $H_2O_2$ and $O_2^{\cdot-}$. ZnPP treatment failed to lower the accumulation of ROS, and dark brown spots and dark blue spots remained higher in the leaves sampled from both assessed rice varieties. There was a reduced accumulation of ROS with the combination of ZnPP and Hemin. Compared to the ZnPP treatment, the number of dark brown spots and dark blue spots decreased in HHZ and XLY900 leaf samples with the SZH treatments.

## The superoxide dismutase, peroxidase, and catalase activity in rice leaves

With the extension of exposure time, the SOD and POD activities in the NaCl treatment of HHZ showed an upward and downward trend, respectively, and CAT activity showed an increased trend compared to CK (Fig. 4). Compared to CK, the SOD, POD, and CAT activities in the NaCl treatment of XLY900 showed an upward trend with the prolonged time of NaCl stress. The SOD and POD activities in NaCl treatment of HHZ reached the maximum at 3 d of NaCl stress, which was significantly increased, by 13.82% and 13.64%, respectively. CAT activities increased by 11.45–21.71% from days 3 to 9 of NaCl stress compared to CK. In comparison to CK, the SOD, POD, and CAT activities of XLY900 under NaCl stress increased by 7.30–26.63%, 6.64–14.53%, and 15.97–24.76%, respectively, from days 3 through to 9. The application of exogenous Hemin boosted the SOD, POD, and CAT activities of SH treatment in the two assessed rice varieties. Compared to NaCl treatment, the SOD, POD, and CAT activities of HHZ with the SH treatment were increased by 4.41–17.66%, 6.48–12.67%, and 6.43–17.33%, respectively, from days 3 through to 9. In comparison to NaCl treatment, the SOD and CAT activities of XLY900 with the SH treatment were increased, by 5.53–27.47% and 10.54–18.12%, from days 3 to 9, respectively, while POD activity increased by 4.53–9.93% except for the day 5. Compared with the NaCl treatment, the ZnPP treatment did not enhance the activities of the assessed enzymes under the applied stress, but instead, was revealed to lower their

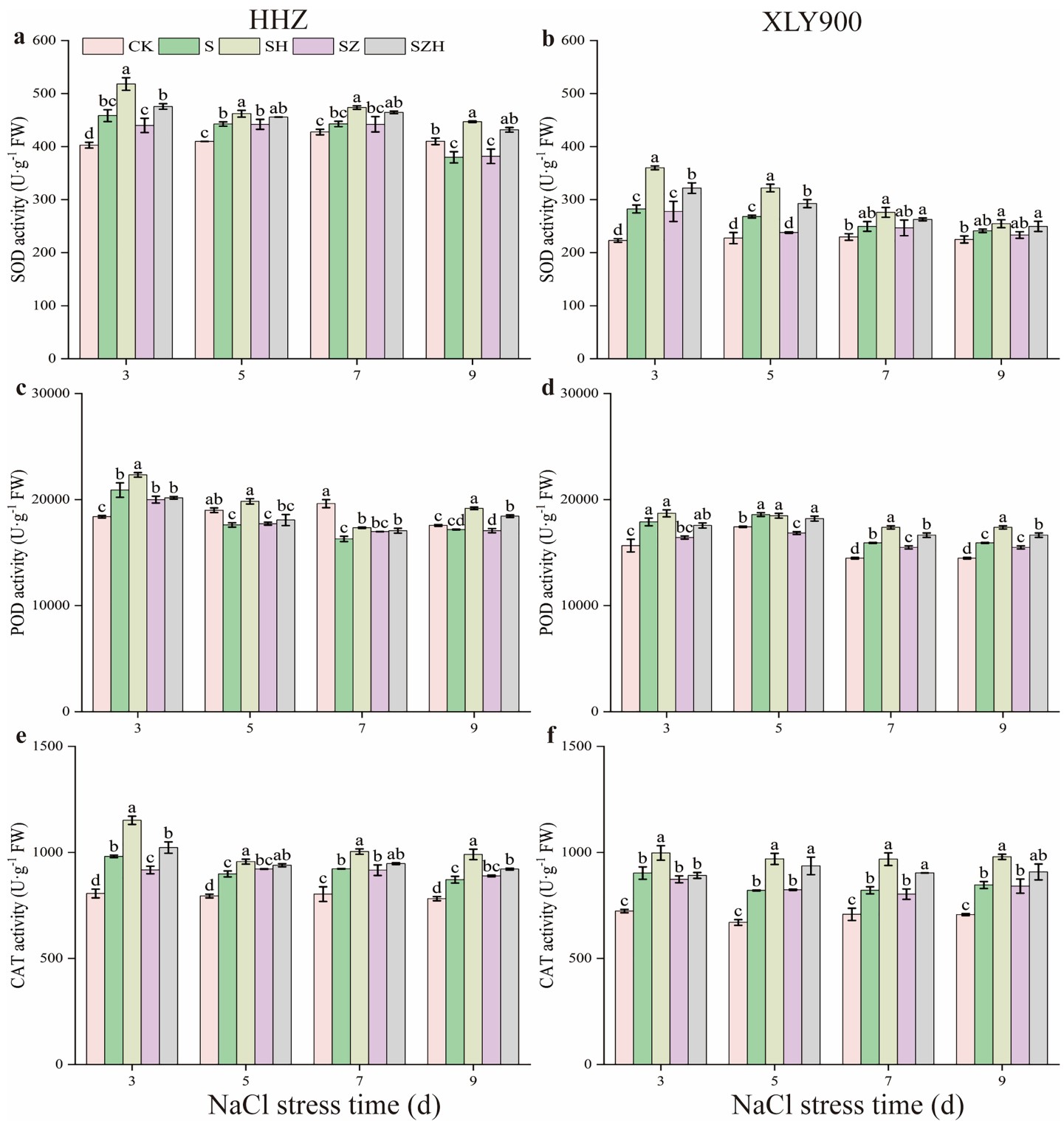

**Figure 4 Effect of Hemin on SOD, POD, and CAT activity of rice seedlings under NaCl stress.** SOD in HHZ (A) and XLY900 (B); POD in HHZ (C) and XLY900 (D) and CAT in HHZ (E) and XLY900 (F). Values are the means ± SD of three replicate samples. Different lowercase letters in the data column indicate significant differences ($p < 0.05$) according to Duncan's test.

activity. For example, compared to NaCl treatments, on day 3, the CAT activity in the SZ treatment of HHZ was significantly decreased by 6.54%; on day 5, the SOD activity in the SZ treatment of XLY900 was significantly reduced by 11.22%. The combination with Hemin relieved the adverse effects of ZnPP and increased the activities of the assessed enzymes. Compared with the SZ treatment, the SZH treatment of HHZ showed SOD activity increased by 3.10–13.12% from days 3 through to 9; POD activity was significantly enhanced by 8.05% on day 9; CAT activity was significantly raised by 11.52% on day 3. Compared with the SZ treatment, the SZH treatment of XLY900 showed SOD activity markedly increased by 15.79% and 22.93% on days 3 and 5, respectively; POD activity significantly enhanced by 7.47–8.07% from days 5 to 9; CAT activity was significantly raised by 13.67% and 12.48% on days 5 and 7, respectively.

## The assessment of the non-enzymatic antioxidants of the AsA-GSH cycle in rice leaves

As the period of NaCl stress was extended, the AsA content decreased, but the DHA and AsA + DHA content increased in the leaves of HHZ and XLY900 (Fig. 5). From days 5 through to 9, compared to CK, the AsA content in the NaCl treatment of HHZ and XLY900 decreased by 2.76–15.52% and 4.62–14.26%, respectively. Compared to CK, the DHA and AsA + DHA content of HHZ under NaCl stress increased by 21.16–60.17% and 4.47–34.18%, respectively. For the XLY900 variety, the assessed parameters increased by 57.73–67.58% and 10.39–32.46%, respectively, from days 3 to 9. The application of exogenous Hemin further increased the AsA content and reduced the accumulation of DHA and AsA + DHA. Compared to the NaCl treatment, the AsA content in the SH treatment of HHZ and XLY900 significantly increased by 4.63–15.54% and 5.46–10.44%, respectively, from days 5 to 9. Compared to NaCl treatment, the DHA and AsA + DHA contents in the SH treatment of HHZ decreased by 15.53–30.23% and 5.06–19.87%, respectively, from day 3 to 9. For the XLY900 variety, the assessed parameters were reduced by 19.87–29.67% and 5.43–12.57% from days 3 to 9. Under NaCl stress, ZnPP treatment mainly raised DHA and AsA + DHA contents in the leaves of the two assessed rice varieties. In comparison to the NaCl treatment, on day 7, the DHA and AsA + DHA contents in the SZ treatment of HHZ were significantly increased by 15.00% and 8.49%; on day 9, the DHA content in the SZ treatment of XLY900 was significantly increased by 8.00%. For the combined treatment of ZnPP and Hemin, the AsA content was higher, and the DHA and AsA + DHA contents were lower in both rice varieties compared to the ZnPP treatment. In comparison to the SZ treatments, on days 5 and 9, the AsA content in the SZH treatment of HHZ was significantly increased by 11.53% and 3.22%, respectively; on days 5 and 7, the AsA content in the SZ treatment of XLY900 was significantly increased by 7.15% and 9.09%, respectively. Compared to the SZ treatment, the DHA and AsA + DHA content in the SZH treatment of HHZ decreased by 12.39–26.77% and 2.81–14.35% from days 3 to 9, respectively. Similarly, the assessed parameters of XLY900 decreased by 8.08–16.27% and 1.72–7.34%, respectively.

Figure 6 showed that as the period of stress exposure was extended, the contents of GSH and GSH + GSSG in NaCl treatment leaves of HHZ and XLY900 decreased, and the GSSG

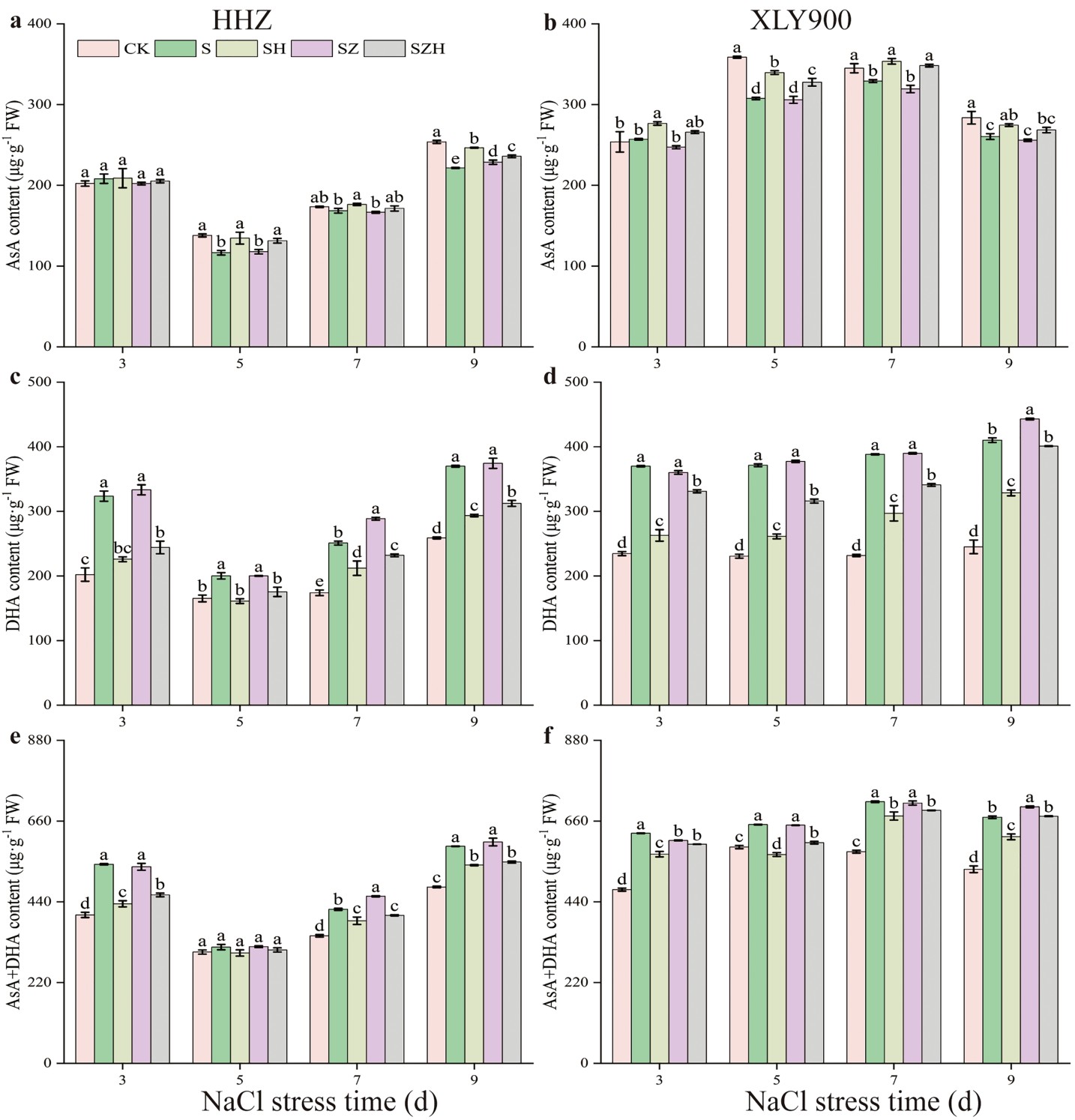

**Figure 5 Effect of Hemin on ascorbic acid content of rice seedlings under NaCl stress.** AsA in HHZ (A) and XLY900 (B); DHA in HHZ (C) and XLY900 (D) and AsA+DHA in HHZ (E) and XLY900 (F). Values are the means ± SD of three replicate samples. Different lowercase letters in the data column indicate significant differences ($p < 0.05$) according to Duncan's test.

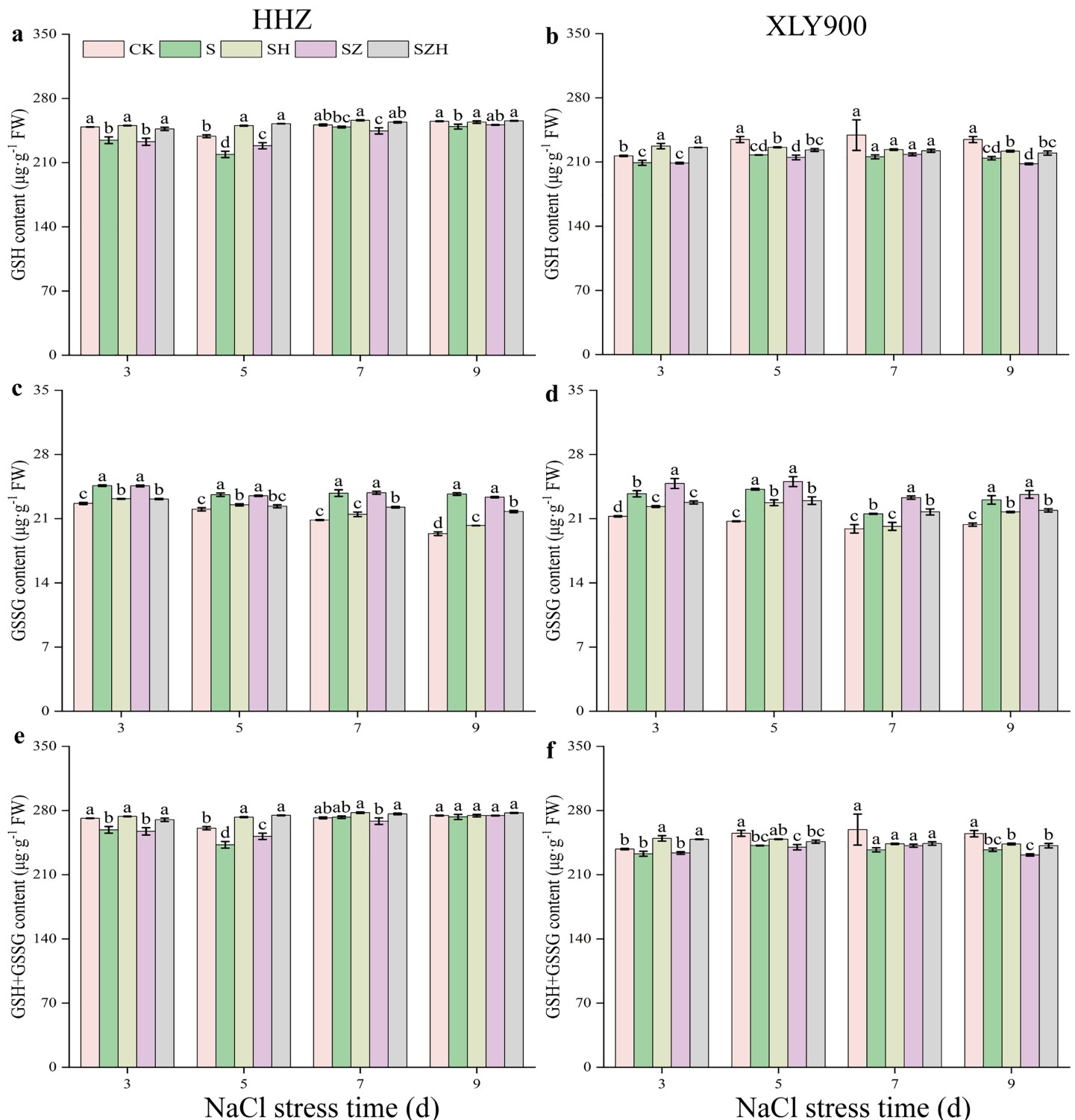

**Figure 6 Effect of Hemin on glutathione content of rice seedlings under NaCl stress.** GSH in HHZ (A) and XLY900 (B); GSSG in HHZ (C) and XLY900 (D) and GSH+GSSG in HHZ (E) and XLY900 (F). Values are the means ± SD of three replicate samples. Different lowercase letters in the data column indicate significant differences ($p < 0.05$) according to Duncan's test.
content in NaCl stressed leaves of HHZ and XLY900 increased. On days 3, 5, and 9, compared to the CK, the GSH content in the NaCl treatment of HHZ significantly decreased by 5.83%, 8.27%, and 2.28%, respectively, and in the XLY900 variety, the GSH content significantly reduced 3.49%, 7.17%, and 8.68%, respectively. From days 3 to 9, compared to the control, the GSSG content in the NaCl treatment of HHZ and XLY900 significantly increased by 7.25–22.36% and 8.20–16.87%, respectively. On days 3 and 5, compared to CK, the GSH + GSSG content in the NaCl treatment of HHZ significantly decreased 4.63% and 6.96%, respectively. Similarly, on days 5 and 9, in the XLY900 variety, the GSH content significantly decreased 5.22% and 6.93%, respectively. The Hemin further increased the contents of GSH and GSH+GSSG and reduced the accumulation of DHA. Compared to the NaCl treatment, the GSH content in the SH treatment of HHZ and XLY900 increased by 1.96–14.31% and 3.60–8.69% from days 3 to 9, respectively. Compared to NaCl treatment, the GSSG content in the SH treatment of HHZ and XLY900 decreased by 4.69–14.52% and 5.74–6.35% from days 3 to 9, respectively. Under NaCl stress, ZnPP treatment mainly raised GSSG content in the leaves. Compared to NaCl treatments, on days 3 and 7, the GSSG content in the SZ treatment of XLY900 significantly increased by 4.71% and 8.24%. In the combination of ZnPP and Hemin, the GSH and GSH + GSSG contents were higher, and the GSSG content was lower in both rice varieties compared to the ZnPP treatment. Compared to the SZ treatment, the GSH content in the SZH treatment of HHZ and XLY900 increased by 1.77–10.55% and 1.80–8.16% from days 3 to 9, respectively. Compared to the SZ treatment, the GSSG content in the SZH treatment of HHZ and XLY900 decreased by 4.90–6.69% and 6.71–8.33% from days 3 to 9, respectively. In comparison to the SZ treatments, on days 3, 5, and 7, the GSSG+GSSG content in the SZH treatment of HHZ was significantly increased by 4.90%, 9.11%, and 2.94%, respectively, and on days 3 and 9, the GSSG content in the SZH treatment of XLY900 was significantly increased by 6.41% and 4.34%, respectively.

**The AsA-GSH cycle enzymatic activities in rice leaves**

As shown in Fig. 7, APX, MDHAR, DHAR, and GR activities were increased along with the period of stress treatment. Compared to the CK, and during the stress period, the activities of the above four enzymes in the NaCl treatment of HHZ were markedly enhanced by 11.00–18.88%, 14.95–54.23%, 23.19–56.82%, and 12.22–27.96% respectively. Similarly, in the XLY900 variety, the assessed parameters were significantly increased 18.82–21.21%, 29.84–51.15%, 19.62–46.87% and 10.48–13.56%, respectively. The use of Hemin further elevated the activities of APX, MDHAR, DHAR, and DHAR. Compared with NaCl treatment, from days 3 to 9, the activities of APX, MDHAR, DHAR, and GR in the SH treatment of HHZ were enhanced by 14.03–25.33%, 19.95–58.63%, 7.10–33.25%, and 8.65–14.11%, while in SH treatment of XLY900 were increased 17.76–26.90%, 11.84–50.44%, 14.67–24.11% and 7.47–12.26%, respectively. However, with ZnPP treatment, the activity of APX, MDHAR, DHAR, and GR was diminished. On day 7, compared to the NaCl treatment, the APX activity of HHZ in the SZ treatment was significantly decreased by 17.03%. On day 9, compared to NaCl treatment, the GR activity of HHZ and XLY900 in the SZ treatment was significantly decreased by 7.14% and 6.46%,

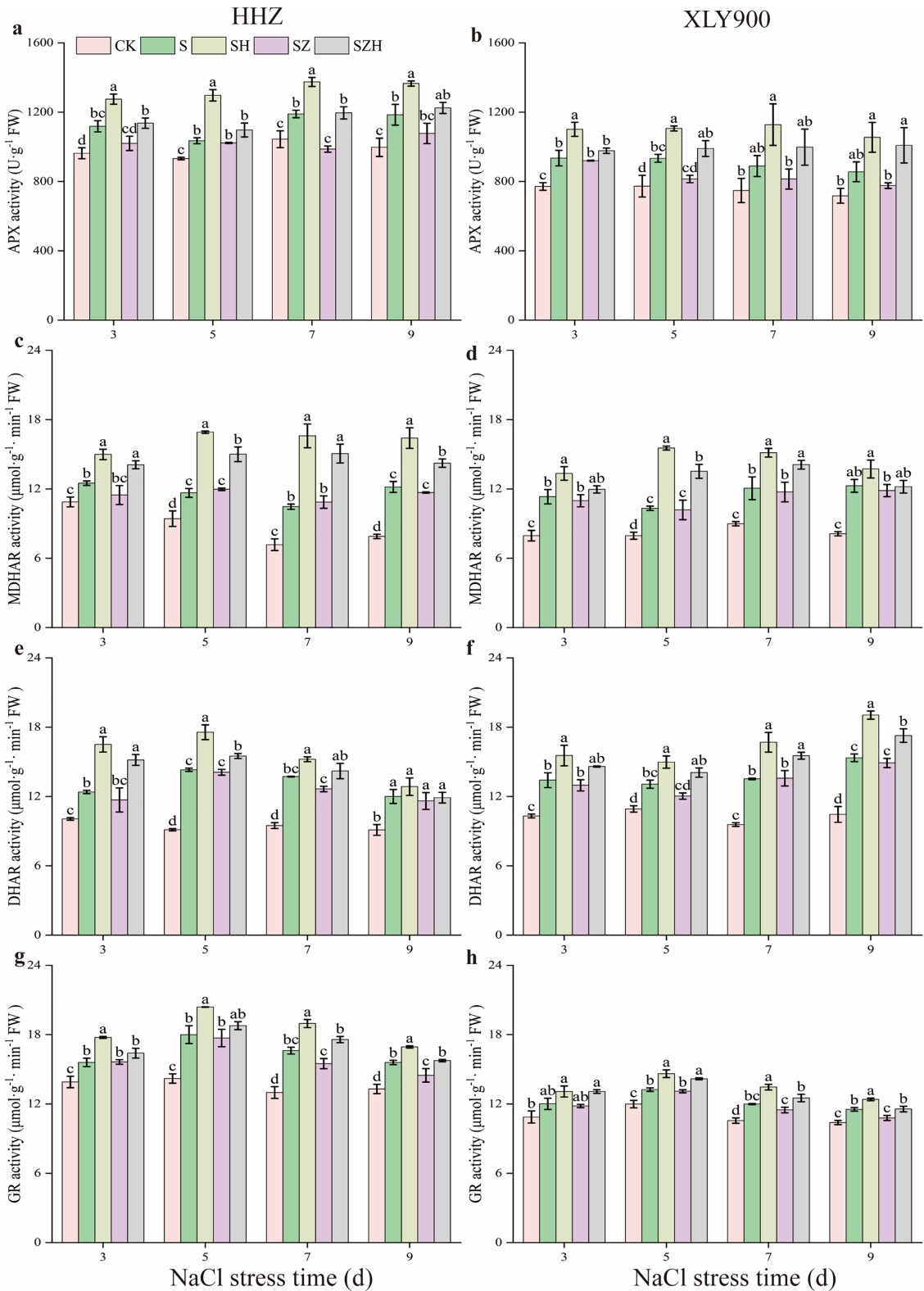

**Figure 7 Effect of Hemin on key enzyme activities in the AsA-GSH defense system of rice seedlings under NaCl stress.** APX in HHZ (A) and XLY900 (B); MDHAR in HHZ (C) and XLY900 (D) DHAR in HHZ (E) and XLY900 (F) and GR in HHZ (G) and XLY900 (H). Values are the means ± SD of three replicate samples. Different lowercase letters in the data column indicate significant differences ($p < 0.05$) according to Duncan's test.

respectively. For the combined treatment of ZnPP and Hemin, the activity of above enzyme was increased. In HHZ with the SZH treatment, the APX activity was significantly increased by 11.41% and 21.15% on days 3 and 7, respectively; the MDHAR activity was markedly increased by 21.78–38.70%, from day 3 to 9; the DHAR activity was dramatically increased by 9.98–29.65%, from days 3 to 7; the GR activity was remarkably increased by 13.47%, and 8.81%, on days 7 and 9, respectively, compared with the SZ treatment. In XLY900 with the SZH treatment, the APX activity significantly increased by 21.60% and 29.99% on days 5 and 9, respectively compared with the SZ treatment. Similarly, the MDHAR activity was markedly increased by 32.81% and 20.13% on days 5 and 7, respectively. Compared with the SZ treatment, the DHAR activity of XLY900 in the SZH treatment was dramatically increased by 14.37–16.89% from days 5 to 9, and the GR activity was remarkably increased by 7.18–9.02% from days 5 to 9.

## The content of osmoregulatory substances in rice leaves

The applied salt stress caused a significant increase in proline content in the leaves of HHZ and XLY900 (Figs. 8A and 8B). Compared to the CK, the proline content of HHZ under NaCl stress was significantly increased by 34.95–65.34% from days 3 to 9. From days 3 to 9, compared to the CK, the proline content of XLY900 with NaCl treatment dramatically increased by 18.95–54.16%. Under NaCl stress, the proline content increased to a greater degree in HHZ than in XLY900. Hemin treatment further enhanced the proline content in the leaves of the two assessed rice varieties. Compared to NaCl treatment, the proline content of HHZ and XLY900 with SH treatment significantly increased by 8.38–27.10% and 15.02–24.35%, respectively, from days 3 to 9. Proline content of rice leaves was not elevated by ZnPP treatment. For example, on day 3, compared to the NaCl treatment, the proline content of XLY900 with the SZ treatment decreased by 8.64%. For the combined treatment of ZnPP and Hemin, the proline content was enhanced. Compared to the SZ treatment, the proline content of HHZ with the SZH treatment had a maximum increase of 26.87% on day 9, and XLY900 with the SZH treatment had a maximum increase of 26.51% on day 7. Compared with the CK, the soluble protein content of HHZ markedly increased in the early stage (on day 3) and then decreased in the later stage (from days 5 to 9) under NaCl stress (Figs. 8C and 8D). The soluble protein content in XLY900 increased during the stress period, with the difference reaching significant levels at all four time points. The foliar application of Hemin enhanced soluble protein content in the leaves of two rice varieties. Compared with the NaCl treatment, soluble protein content in the SH treatment of HHZ noticeably increased by 2.75% on day 9. While it in the SH treatment of XLY900 significantly elevated by 3.93% and 1.17% on days 5 and 7, respectively. Spraying ZnPP did not increase soluble protein content. For example, on day 3, soluble protein content significantly decreased by 3.20% in the SZ treatment of XLY900 compared with the NaCl treatment. When ZnPP was combined with Hemin, soluble protein content was enhanced. For example, on day 3, the soluble protein content of HHZ and XLY900 in the SZH treatment was increased by 3.02% and 3.21%, respectively.

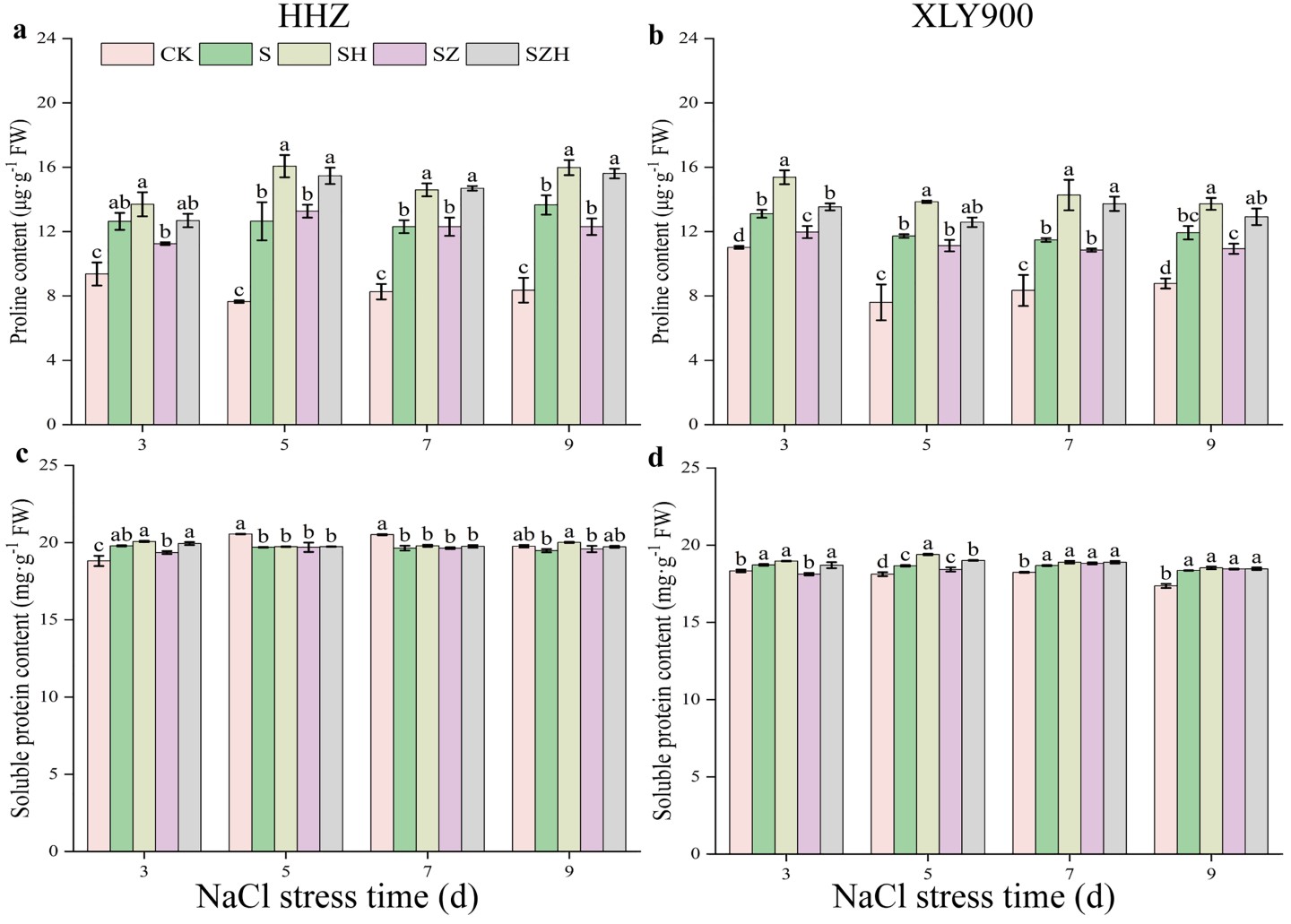

**Figure 8 Effect of Hemin on osmoregulatory substances of rice seedlings under NaCl stress.** Proline content in HHZ (A) and XLY900 (B); soluble protein content in HHZ (C) and XLY900 (D). Values are the means ± SD of three replicate samples. Different lowercase letters in the data column indicate significant differences ($p < 0.05$) according to Duncan's test.

## DISCUSSION

Globally, salt stress is the most prevalent abiotic stress that limits crop growth and development. Studies have shown that salt stress impedes the growth of several crops, such as wheat (*Ashraf et al., 2023*), sorghum (*Liu et al., 2023*), and soybean (*Feng et al., 2021*). Excessive salt interferes with normal biological and physiological processes to negatively impact plant growth (*Talubaghi et al., 2022*), such as reduced plant height, narrowed stem base width, and diminished biomass. The results obtained from the experimentation performed in this study were similar to those reported previously (*Ashraf et al., 2023*; *Liu et al., 2023*; *Feng et al., 2021*; *Talubaghi et al., 2022*). More specifically, under salt stress, the seedling growth of both HHZ and XLY900 was significantly inhibited, and all the morphological indexes were decreased (Tables 1 and 2). Foliar application of Hemin positively regulated various morphological indicators and promoted shoot growth and biomass accumulation in rice seedlings. *Liu et al. (2021)* showed that Hemin improved the

growth of maize seedlings and increased biomass accumulation under drought stress. Furthermore, Hemin was degraded in plants to produce CO, which alleviated the inhibition of wheat growth by NaCl stress (*Ling et al., 2009*). Exogenous ZnPP was unable to promote rice growth under salt stress in this study, which is consistent with a previous report (*Cao et al., 2011*).

ROS can be used at low concentrations as a secondary messenger or signaling molecule (*Antoniou et al., 2016*). Plants generate and remove ROS in dynamic balances under normal growth conditions. Under abiotic stress conditions, ROS levels surge and are destructive, leading to changes in the structure of DNA, proteins, and enzymes, ultimately resulting in programmed cell death (*Gill & Tuteja, 2010*; *Singh et al., 2019*). MDA is one of the membrane lipid peroxidation products whose content can reflect the level of ROS and the degree of membrane lipid peroxidation. EL can evaluate cell membrane permeability. The higher EL value indicates greater damage to the cell membrane (*Ben Youssef et al., 2021*). In this experiment, the findings showed that salt stress caused higher EL, increased MDA and $H_2O_2$ contents in the two assessed rice varieties and that the results were positively correlated with stress duration (Fig. 2). Compared with XLY900, HHZ had a much greater increase in the above three indexes, indicating that HHZ was more sensitive to the NaCl stress. This was similar to a previous study (*Chen et al., 2022*). The localization of $H_2O_2$ and $O_2\cdot^-$ in leaves was measured by histochemical methods. Salt stress induced the accumulation of $H_2O_2$ and $O_2\cdot^-$ in the leaves of HHZ and XLY900 compared with the CK (Fig. 3). This conformed with the finding of *Jabeen et al. (2020)* who studied cultivated rice under salt stress previously. Previous studies have shown that Hemin can mitigate the damage caused by stress in plants, reducing ROS accumulation, MDA content, and cell membrane permeability (*Chen et al., 2009*; *Cui et al., 2012*). In this experiment, exogenous Hemin effectively diminished EL, $H_2O_2$, and MDA contents (Fig. 2), reduced $H_2O_2$ and $O_2\cdot^-$ accumulation (Fig. 3) and alleviated the damage of salt stress to the cell membranes. Exogenous ZnPP could not scavenge excess ROS and maintain cell membrane stability. When ZnPP was combined with Hemin, it scavenged part of the ROS and alleviated oxidative damage, this finding was in agreement with the previous finding by *Zhang et al. (2012)*.

Facing stress, plants activate antioxidant defense systems to minimize damage caused by oxidative stress. Among them, antioxidant enzymes mainly include SOD, CAT, and POD. SOD represents the first barrier for plants to resist ROS damage caused by abiotic stresses and catalyzes the transformation of $O_2\cdot^-$ to $O_2$ (*Karuppanapandian & Kim, 2013*). CAT eliminates $H_2O_2$ with minimal energy consumption and very high conversion rates for large-scale scavenging of ROS (*Zamocky et al., 2012*). POD has a strong affinity for $H_2O_2$ and is used for the fine-tuning modulation of $H_2O_2$ (*Abogadallah, 2010*). In this study, compared with the CK, SOD, and POD activities of NaCl treatment in HHZ were firstly increased and then decreased, and CAT activity was increased (Figs. 4A, 4C, and 4E); but SOD, POD, and CAT activities of NaCl treatment in XLY900 were increased with stress duration (Figs. 4B, 4D, and 4F). This indicated that rice eliminated ROS in the short term of salt stress by increasing the activity of antioxidant enzymes; while in the long term, rice accumulated more ROS, which could not be scavenged in time by antioxidant enzymes.

The results obtained from the experimentation performed in this study were similar to those reported previously by others (*Vaidyanathan et al., 2003*; *Seckin, Sekmen & Turkan, 2009*; *Kumari et al., 2023*). Foliar application of Hemin enhanced SOD, POD, and CAT activities in leaves of the two assessed rice varieties under assessment when exposed to salt stress (Fig. 4). This demonstrated that exogenous Hemin stimulated the antioxidant enzyme system in rice and facilitated the increase of enzyme activities, which avoided oxidative damage and ensured normal plant growth. The inhibitor ZnPP could not increase the activities of antioxidant enzymes or even inhibit them. ZnPP combined with Hemin mitigated the inhibitory effect caused by ZnPP (Fig. 4). Based on a previous study (*Zhang et al., 2012*), it is hypothesized that Hemin enhances antioxidant enzyme activity in rice leaves by promoting heme oxygenase (HO) expression thereby increasing the antioxidant enzyme activity. ZnPP acts as an inhibitor of HO, hindering expression which restricts the increase in antioxidant enzyme activity.

The AsA-GSH cycle is an essential ROS scavenging mechanism. It mainly consists of the antioxidant enzymes APX, MDHAR, DHAR, and GR and the nonenzymatic antioxidants AsA and GSH, which can alleviate the oxidative damage caused by salt stress (*Wang et al., 2022*). As part of the cycling process, under APX catalysis, AsA converts $H_2O_2$ to $H_2O$, which is then oxidized to DHA. DHA is converted to AsA by a reduction oxidation reaction with MDHAR or to DHA by a non-enzymatic disproportionation reaction. DHAR catalyzes DHA and GSH to produce AsA and GSSG, while GSSG can be restored to GSH by GR (*Nahar et al., 2015*; *Tan et al., 2022*). AsA and GSH act as nonenzymatic antioxidants and assist other antioxidant enzymes in scavenging ROS. In this study, salt stress decreased AsA content and increased DHA and AsA + DHA contents in rice leaves (Fig. 5), indicating that APX activity enhancement decreased the AsA content. Foliar application of exogenous Hemin significantly improved AsA content and diminished DHA and AsA + DHA contents. This finding suggests that the increase in MDHAR and DHAR activities caused an increase in AsA content and a decrease in DHA content. Under salt stress, GSH and GSH + GSSG contents decreased, and GSSG content increased, while exogenous Hemin treatment increased GSH and GSH + GSSG contents and decreased GSSG content in rice leaves (Fig. 6). This showed that the enhanced GR activity facilitated the conversion of GSSG to GSH and maintained a high level of GSH in its reduced state, which was in agreement with the research of *Piao et al. (2022)*. Together, these findings indicate that Hemin improves cellular reduction ability at a high level to protect against oxidative damage. In addition, in this experiment, salt stress increased the activity of APX, MDHAR, DHAR, and GR in the two assessed rice varieties compared with the control (Fig. 7). This finding indicates that salinity stress increases the $H_2O_2$ content of rice leaves, which prompts APX to accelerate the scavenging of $H_2O_2$, while the increased activities of MDHAR, DHAR and GR are beneficial to the resistance of a leaf to oxidative damage, which is a stress response to excess $H_2O_2$. Foliar application of exogenous Hemin further induced the activities of APX, MDHAR, DHAR, and GR (Fig. 7). A previous study suggested that this might be possible by upregulating the transcription of genes for enzymes related to the metabolism of the degradation products CO and GSH, which could increase the enzyme activity to help plants mitigate the oxidative damage caused by the

stress (*Zhang et al., 2016*). ZnPP could not be degraded to $CO_2$ in plants and reduce endogenous $CO_2$ production by blocking HO expression. Thus, it could not enhance the activities of MDHAR, DHAR, and GR under salt stress. Moreover, Hemin induced *HO* expression and enhanced gene expression of critical enzymes in the AsA-GSH cycle, while ZnPP prevented *HO* expression and even strengthened the inhibitory effect of NaCl stress on the AsA-GSH cycle in rice seedlings (*Cui et al., 2012*). These results reflect that Hemin improves the efficiency of ROS scavenging in rice, which maintains cell membrane stability and enhanced the resistance of rice.

Although saline soil contains water, plants cannot absorb the water, mainly because the soil has a high level of ion that increases the osmotic pressure of the external environment, which prevents plant cells from absorbing water, and this elevated soil ion content can even lead to the loss of water from plant root cells. Therefore, plants ensure water absorption by increasing their production of osmoregulatory substances and decreasing the difference in osmotic potential between their internal and external environments. The important osmoregulatory substances (soluble proteins and proline) have different physiological functions in maintaining osmotic balance in plants. Soluble proteins can assist in binding water molecules in plant cells and maintain the stability of the cell structure (*Hao et al., 2021*). Proline is a potential non-enzymatic antioxidant that scavengers of single-linear oxygen molecules and hydroxyl radicals. Thus, proline prevents lipid peroxidation of cell membranes and avoids exposure of plants to ROS-induced oxidative damage (*Szabados & Savoure, 2010*). In this study, we found that with the increased period of NaCl stress exposure, soluble protein content initially increased and then decreased in HHZ, while it continued to increase in the XLY900 variety (Fig. 7). A previous study has shown that salt stress disrupts the protein synthesis pathway during its later stages, and in combination, accelerates the rate of protein catabolism, to generate large amounts of amino acids, and ultimately reduces protein content (*Alisofi, Einalia & Sangtarash, 2020*). This could be why there is a decrease in soluble protein content in HHZ. The soluble protein content in XLY900 was enhanced to relieve the difference in osmotic potential. The two assessed rice varieties exposed to salt stress had significantly increased proline content. Compared with the XLY900 variety, salt stress caused the HHZ variety to produce much more proline (Figs. 8C and 8D). This was similar to the result of *Gao et al. (2016)*, in which the salt-sensitive variety had high proline content when exposed to stress. Foliar application of exogenous Hemin promoted the accumulation of osmoregulatory substances in rice leaves, significantly increasing soluble protein and proline contents. However, in the ZnPP treatment, the content of osmoregulatory substances was reduced instead of increased, with similar observations previously reported by *Zhao et al. (2022)*. Together, these results indicate that Hemin induces a large accumulation of proline and soluble proteins, which is beneficial for the absorption of water and the maintenance of cellular osmotic pressure in rice leaves under salt stress.

## CONCLUSIONS

During the seedling stage, the activity of antioxidant enzymes and the content of non-enzymatic antioxidants initially increased in response to salt stress. This response

effectively countered the accumulation of ROS induced by the stress. However, with prolonged exposure to stress, the activity of enzymes continued to increase while the content of the antioxidants decreased, failing to alleviate the stress promptly.

The accumulated ROS and membrane lipid peroxides exacerbated the damage caused by the imposed stress, eventually leading to a decrease in growth. The application of Hemin enhanced the activity of antioxidant enzymes and elevated the content of non-enzymatic antioxidants, which contributed to an overall improvement in the antioxidant capacity of rice, resulting in a reduction of membrane lipid peroxidation. This, in turn, ensured the continued functionality of the AsA-GSH cycle to enhance rice's resistance to the imposed stress.

### Funding

This work was supported by the Special Projects in Key Areas of Ordinary Colleges of the Educational Commission of Guangdong Province (2021ZDZX4027), the Innovation Team Project of ordinary colleges of the Educational Commission of Guangdong Province (2021KCXTD011), the Zhanjiang Science and Technology Bureau (2022A01016), the Research start-up project of Guangdong Ocean University (R20046), and the Research start-up project of Guangdong Ocean University (060302052012). The funders had no role in study design, data collection and analysis, decision to publish, or preparation of the manuscript.

### Grant Disclosures

The following grant information was disclosed by the authors:
Special Projects in Key Areas of Ordinary Colleges of the Educational Commission of Guangdong Province: 2021ZDZX4027.
Innovation Team Project of ordinary colleges of the Educational Commission of Guangdong Province: 2021KCXTD011.
Zhanjiang Science and Technology Bureau: 2022A01016.
Research start-up project of Guangdong Ocean University: R20046.
Research start-up project of Guangdong Ocean University: 060302052012.

### Competing Interests

The authors declare that they have no competing interests.

### Author Contributions

- Fengyan Meng conceived and designed the experiments, authored or reviewed drafts of the article, and approved the final draft.
- Naijie Feng conceived and designed the experiments, authored or reviewed drafts of the article, and approved the final draft.
- Dianfeng Zheng conceived and designed the experiments, authored or reviewed drafts of the article, and approved the final draft.

- Meiling Liu performed the experiments, prepared figures and/or tables, and approved the final draft.
- Hang Zhou performed the experiments, prepared figures and/or tables, and approved the final draft.
- Rongjun Zhang analyzed the data, prepared figures and/or tables, and approved the final draft.
- XiXin Huang analyzed the data, prepared figures and/or tables, and approved the final draft.
- Anqi Huang analyzed the data, prepared figures and/or tables, and approved the final draft.

## Data Availability

The raw measurements are available in the Supplemental Files.

## Supplemental Information

Supplemental information for this article can be found online at http://dx.doi.org/10.7717/peerj.17219#supplemental-information.

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
