# Peer review of "Exogenous Hemin enhances the antioxidant defense system of rice by regulating the AsA-GSH cycle under NaCl stress"

_PeerJ, doi:10.7717/peerj.17219_

## Round 0.1 · original submission · Major Revisions

Dear authors,
Post receiving the reviewer comments from two experts in the field, I have also assessed your article.

I share the opinion of Reviewer #1 that your submitted manuscript requires MAJOR REVISION before it can be considered further for publication in PeerJ.

Editor comments:

Overall, the English language used in the manuscript is of a high standard. However, please thoroughly check and revise the wording of the Results section as many sentences are quite repetitive. Please be more specific when discussing your results.

I request more information on the following outstanding questions:

- Why were the two cultivars selected for study inclusion over other rice cultivars that would have also been available for use by the authors?

- Outline why the specific growth stage of the rice plants was selected for assessment (that is: why the 3 leaf / one heart stage for spray application)?

- Why was the specific concentration of NaCl applied / how was this concentration of NaCl in the foliar spray selected?

- How was the salt concentration of the soil accurately determined?

- Why sample at days 3, 5, 7, 9 and 50 after treatment application? What was the basis of selecting these time points for sampling?

- Why are whole plant images not included in the manuscript - even as a Supplemental Figure? This is a significant oversight by the authors for such a study.

Authors are to also address the Reviewer #1 comments:

- Please address the question on the inclusion of the ZnPP data.

- Please outline if salt stress was to be applied prior to the Hemin - what would the observed effect be?

- Please comment of the accuracy of some of the data presented in Table 1.

Authors are to also address the Reviewer #2 comments:

- Please address why the study data was analysed in the way it was?

- Please try to simplify the degree of data presented in Table 1

- Why were the two rice cultivars selected for study inclusion?

Reviewer 1 ·

Basic reporting

no comment

Experimental design

Major point:
For all the experiments, ZnPP data is included. ZnPP acts as a competitive inhibitor of HO-1. HO-1 improves salt resistance, and Hemin induces the expression of HO-1, increasing the activity of HO-1. Why did the authors treat rice with ZnPP under stress? Was it to make rice grow worse? It seems that the ZnPP data is not helpful to the conclusion. What are the authors' thoughts on this? Does this paper really need ZnPP data?

Minor points:
1. In the study, rice was exposed to salt after Hemin spraying. What happens if salt treatment comes first? Does Hemin still function as an antioxidant?
2. Does Hemin affect rice growth if the plant was treated with Hemin only?
3. How many plants were measured in each replicate?
4. On line 139, please provide details of the method.
5. On line 202, it is mentioned that the Duncan test was used. Is it one-way ANOVA followed by Duncan's test?

Validity of the findings

In Table 1, please verify the accuracy of the data.

Stem diameter
XLY900 SZ SZH 9d

Shoot fresh weight (g)
HHZ SZ SZH 9d

Additional comments

The authors have submitted a manuscript to study the morphology and physiology of rice under salt stress. They found that Hemin plays a role in enhancing the antioxidant defense system. This study provides information for the potential application of Hemin in rice cultivation under salt stress. The conclusions, at least for the Hemin part, are supported by the data. However, before granting publication, there are some issues the authors may need to address.

·

Basic reporting

The manuscript entitled "Exogenous Hemin enhances the antioxidant defense system of rice by regulating the AsA-GSH cycle under NaCl stress" describes the role of hemin to mitigate salt induced damages in rice plants. The experiment was executed nicely and the results are good. The preparation of manuscript was properly done, and well written, but the manuscript needs some corrections.
 In abstract, should include the full form of ZnPP.
 Keywords need to be rearranged in alphabetical order.
 Authors should care about the scientific name, in some places such as line no. 90, 91 were not properly written. Please check whole manuscript including references.

Experimental design

Why did the author select two rice varieties for this study? It should shortly described in the methodology.
 The authors used two rice varieties and data observed four different growing times. So, data should be analyzed as a factorial design to observe the varietal difference.

Validity of the findings

The presentation of data in Table 1 is very complex; author should revise it to simplify this.
 To observe the phenotypic differences, plant pictures should be included.
 Please check the parameter name in Fig. 2. For all parameters observed four different times, but in the case of Fig. 2 did not mention the time. The author should check and mention the time.

Additional comments

What are the possible mechanisms of hemin to improve antioxidant defense mechanisms?

---

## Round 0.2 · Major Revisions

Dear Authors,

Please very carefully review the annotated PDF that I have attached to this decision letter which clearly outlines why I have decided that your revised manuscript version still requires considerable improvement prior to being considered further for publication.

I acknowledge that you have adequately addressed the comments / concerns raised by the reviewers post their review of your originally submitted manuscript in the revised manuscript version, however, much work is still required to further improve the standard of your study.

Please make all corrections that I have highlighted on the annotated PDF. In addition to this, I strongly recommend a through revision of the text of each of the individual sections of the manuscript to ensure that each section is of a high standard.

Once you have made my requested changes, please resubmit a fully revised version of your manuscript for my further consideration.

Kind regards,
Andrew

Reviewer 1 ·

Basic reporting

All concerns have been addressed

Experimental design

All concerns have been addressed

Validity of the findings

All concerns have been addressed

Additional comments

All concerns have been addressed

·

Basic reporting

no comment

Experimental design

no comment

Validity of the findings

no comment

---

## Round 0.3 · Minor Revisions

Dear authors,

I have asked you previously to please take care (extra care) when completing each of the requested changes to the current version of your manuscript.

You have failed to do so to an acceptable standard on many occasions throughout the text of your manuscript in preparing this revised manuscript version. Please do take care making the requested changes to your manuscript.

All requested changes to the text of your manuscript are required for me to consider your manuscript further. Please do not rush this process, this will ensure that all of my concerns which remain regarding your manuscript are addressed accordingly.

Please make all changes to the text I have highlighted in the attached and annotated PDF.

Thank you,
Andrew Eamens

---

## Round 0.4 · Minor Revisions

Dear authors,

I am once once again asking you to take care when revising your manuscript.

I do not want to have to review this manuscript again, so please do address all remaining problems with the manuscript.

Thank you,
Andrew

---

## Round 0.5 · Minor Revisions

Dear authors,

The two main issues remaining in the current version of your manuscript are:

1. Please once again very carefully the text of the entire manuscript (all sections must be checked and corrected) as many grammatical errors, repeated words, and other inconsistencies (e.g., 5% versus 5.0%, etc) still remain. These issues must be addressed prior to acceptance of the study. I am not once again providing an annotated version of your manuscript, as it is up to the authorship team to identify and change the text of the manuscript as required.

2. Please ensure that the statistical analyses are correct for this study. On many instances, increases or decreases of 1-5% or around 10% are stated to be statistically significant differences. I find it very hard to believe that such minor changes in values represent statistical differences.

If such changes in values are determined to indeed not be statistically different then the descriptive text describing each set of results, must be changed accordingly.

Kind regards,
Andrew

---

## Round 0.6 · Minor Revisions

Dear authors,

Additional changes (text editing) are required to further improve the standard of your manuscript.

This represents the final round of editing which will be completed for your manuscript, so please address / correct all changes that I have requested in the latest round of review.

If the requested changes are not made to this version of your manuscript, then this will be viewed as a lack of willingness to improve your manuscript any future, and as such, I will have to reconsider the status of the manuscript.

Kind regards, Andrew

---

## Round 0.7 · accepted · Accept

Dear authors,

Thank you kindly for working through the multiple rounds of revision on your submitted manuscript. These multiple rounds of revision have resulted in the production of a high quality manuscript suitable for publication in PeerJ.

I would however recommend a final read-through of the manuscript text by your authorship team prior to submission of a finalised version for publication.
And finally, congratulations on producing a high quality and highly interesting piece of research.

All the very best,
Andrew Eamens